# Promoting leisure functions through setting creative linguistic landscapes in recreational zones

Kun Sun, Xiaoli Tian*, Jing Xia, Qing Li, Bing Hou

School of Tourism & Culinary, Yangzhou University, Yangzhou, PR China

* tianxiaoli0305@163.com

**Data Availability Statement:** All files related to the study are available from Harvard Dataverse https://doi.org/10.7910/DVN/VGDVRF.

## Abstract

Using creativity to promote recreational services is crucial. Accordingly, creative linguistic landscapes (CLLs) are being used to improve visitors' experiences in some recreational zones. However, relevant research is still in its early stages. Therefore, this study was conducted. It summarized the leisure function categories and function evaluation indicators of CLLs in recreational zones respectively based on image materials and related online reviews. The leisure function outcomes of all CLL types were ranked using the fuzzy PROMETHEE method; based on this ranking, a CLL configuration optimization mode was suggested. The findings reveal the following. (1) Currently, there are mainly nine leisure function types of CLL in practice, although the type structure is severely imbalanced; there are 12 primary corresponding function evaluation indicators, although each of them draws significantly different attention. (2) There are notable variations among the outcomes of different types of functions of CLL: mood adjustment is the most advantageous function of CLL for leisure services, followed by emotional guidance and cognitive building functions; (3) According to the study findings, in the configuration of CLL, which aims at leisure function optimization, the "function focusing and coordinating mode (the superior functions of CLL are focused on and its various functions are coordinated)" should be adopted. The results provide meaningful lessons for the establishment of rational and effective CLL in recreational zones.

## 1. Introduction

Language plays an important part in the operations of tourist zones to yield the leisure functions [1], and language signage constitutes an essential component of tourist infrastructure. The language exhibition is deemed as linguistic landscape (LL) [2]. Actually, no tourism operator does not use linguistic landscape [3]; The linguistic landscape always first come into tourists' views, displaying essential information [4]. The leisure functions of LLs in recreational zones were revealed sufficiently by relevant researches, for example, changing the spatial aura [5], reflecting the ongoing development of leisure culture, setting the trend for leisure [6], symbolizing tourism value [7], among others.

**Funding:** For this research, BH was supported by the Art Project of National Social Science Fund of China (award number 23BH164). No additional external funding was received for this study. The funder had no role in study design, data collection and analysis, decision to publish, or preparation of the manuscript.

**Competing interests:** The authors have declared that no competing interests exist.

The LL elevates language as the semiotic resource [8]. The creative words are special semiotic resources, contribute to accomplish a great deal of meaning-making [5]. A viewed language carrier with creative words, graphical language, or shape, is defined as creative linguistic landscape (CLL) in this study. Because creativity is a key tool for recreational places that wish to innovate their service functions and improve available experiences [9], which can enhance their attractiveness [10], CLL is also attached more attention by tourism operators, which has greater potential to provide visitors with a better experience. In some recreational zones, a distinctive linguistic landscape is formed through creative and characteristic signage with novel text content [11]. At some sites, setting up a CLL has been a crucial strategy for attracting visitors [12], and a large number of CLLs have emerged in some recreational zones [13]. CLLs can further enrich the medium by which visitors perceive and interact with these places in a more interesting and unique way [14]; for example, a linguistic landscape with the novel text "*let us brew the blissful Luzhou (a place famous for producing liquor)*" can act as a medium by which visitors become more active and interested in perceiving and experiencing the brewing culture of Luzhou. Nowadays, highly creative content is more likely than conventional content to be shared on social media [15]. This makes some CLLs "internet-famous" and makes them a powerful tool for increasing visitor flow in recreational zones. The installation of CLLs has significantly increased the number of people visiting relevant destinations [16].

As a result, the installation of CLLs has become a significant phenomenon in many recreational zones. In view of this, some questions need to be addressed for better setting CLLs. (1) What functions do CLLs have is the first critical question to be addressed to provide basis for installing them according to practical requirements. As mentioned above, the leisure functions of the LL are fully discussed, however, those of the CLL haven't been attached enough research attention yet. Therefore, this study devotes to bridge this gap. (2) Operators intend to choose effective CLL to boost tourist numbers [17], and this depends on the functional evaluation of different CLLs. In addition, despite many benefits, some CLLs have also drawn significant criticism. Some people think that they interfere with conventional LLs and other sceneries; Emotional language is becoming popular in some recreational zones, however, it sometime makes people feel awkward; Some CLLs also cause misunderstandings [18]. So, the positive and negative functions of CLLs require to be evaluated comprehensively, leading to their positive/negative functions being released/controlled. Therefore, how to evaluate CLL functions roundly is the second critical question, which hasn't been fully discussed in existed researches. Addressing this question is also this study objective. (3) Because many operators are getting down to using LLs to attract more visitors [19], and shape the competitive image of place [6], tourists may encounter a bombardment of language carriers in some recreational zones [20]. An avalanche of language signs may bring confusions to tourists [3]. So, how to reasonably and effectively configure CLLs to make them function better is the third critical question. Likewise, this hasn't aroused enough researches yet. This study also attempts to address this question to provide a reference for CLLs installation.

## 2. Literature review

### 2.1 Linguistic landscape functions in recreational zones

The LL is the most significant component of a recreational zone, and contributes to tourist experience; even in a historic battleground, there is nothing to see besides the LL [3]. Recreational service operations rely heavily on LLs, for interpretation, instruction, regulation, stimulation [21], promotion, information provision [1], environmental education [22], among others. In practice, LLs can act as a form of unobjectionable soft governance [23].

Firstly, The LL plays crucial functions in representing the recreational zone. It is always used to show the spatial indexicality and iconicity [24], visualize the development effort there [25], shape the place image [26] and a clear theme for the attraction [27], demonstrate the identity of the zone [28], share related culture and spread knowledge [29], among others. These will increase the significance of the zone [3]. Secondly, The LL has significant impact on tourists' senses. Well-designed LLs are emotionally and intellectually stimulating [30], can promote visitors' enthusiasm for nature [27], make recreational zones more aesthetically pleasing [31], facilitate their historical authentic senses [32], raise their awareness and senses of meaning [33], cater to their taste and manifest exoticism [26], promote their national pride and identity [24], and others. The LL is the medium par excellence for reflecting and transmitting imagination and imaginaries [34], contributes to create a sense of identity for visitors [35]. Thirdly, the LL can be used to guide and regulate their behavior [36]. LLs are frequently used to inform visitors of rules and warnings [37], direct them to efficient travel trails [38], facilitate their learning [33] and communication with a place [39], assist them in wayfinding behavior [15]. LLs can also provide the crucial medium for interactions between hosts and guests [40], promote tourists' purchase of specialty goods [41], etc.

In recreational zones, Unique LLs can serve as an important symbolic resource and visual consumption content promoting visitors' experiences [42], which are always shaped through signage in recreational zones [11]. However, in some zones, the LL does not meet tourists' needs effectively [38]. CLLs can meet some new needs of visitors, such as for emotional exchange [6], catering to wishes [33], mood regulation, and so on. However, research on satisfying visitors' needs and optimizing their experiences through CLLs still remains scarce.

## 2.2 Linguistic landscape construction in recreational zones

In recreational zones, mainly for profit-making purposes [4], operators always set up LLs to draw in tourist [19], communicate with visitors [43], reduce manpower expense through the guidance functions of LLs [5]; sometime, it is also established for local pride [4], or others. Relevant researches identified some problems in setting up LLs, for example, in some tourism signs, words were simply stacked, being lack of linguistic aesthetic; many linguistic landscapes didn't embody the local history and culture [44]; some private owners pay more attention on the modernity in setting up LL, instead of on traditional heritage [19], among others.

More researches have made generative suggestions for the optimal construction of LLs. (1) To coordinate different LLs, the operator should make them similar along the same route to minimize visual pollution from various LL types [45]. Unity, rhythm, scale, color, and visual perception continuity should be considered when designing and setting LLs in recreational zones [31]. (2) LL construction should achieve a balance between the operational needs of the place and visitor demand. The operator should pay attention to the reading appeal of words from the perspective of different categories of visitors. For instance, children's height should be considered for educational LLs [46]. Language used should be concise and easily understood regardless of age or physical condition [47]. (3) To improve the readability of an LL, contrasting its color with the color of the surroundings is better [48]; diagrams can be used to make an LL more noticeable [47], and more creative LL styles should be developed to better catch visitors' eyes, engage readers, and occasionally make them smile [37]. The language used need to be clear and informative [17]. Large and well-spaced text should be used to offer tourists convenience for reading; safety signage should represent short and familiar words, and be separated from other information [49]. It is better to place LLs with more words in rest areas than in fast-moving high-traffic areas, so that visitors have time to read the words [50]. (4) For quality LLs, the content should be thematic [46], and a provocative title is important [51].

Designers can add thought-provoking content to LLs to cause visitors to think about specific matters such as the relationship between people and the environment [52]. Sometime, the music, film can be associated with the linguistic landscape to create stronger impact on tourists [3]. Special attention should be paid to LLs with evidence-based content, which can affect visitors more [37]. (5) For tourists' experiences, previous research emphasizes that the autochthonous language should be valued to cause tourists to experience the genuine [19]; and minority language can be used to promote cross-cultural experience [41]. In addition, paper, cloth, metal and wooden can be used as carriers of LLs to diversify tourists' experience.

Some researches also reveal the distinctions of the LL setting in China. Comparing with English-speaking countries, in China, more types of language signs are used in recreational zones; especially, the language used on bottom-top signage is more diverse, although the top-bottom LL is dominant [53]; meanwhile, the font types with obvious distinction are very rich; then, the operators show strong subjectivity in choosing the language type, for example, some use Korean in the signage, but others use German or Malay, among others [43]. Notably, the English letters can be changed more flexibly in shape and size, so, the effect of English signage can be improved through the interesting and artistical design of the letter shape and size, however, Chinese characters are different, the effect of Chinese signage is always improved through enhancing its capacity to express the emotion, symbolic meaning, imagery and spirit [54].

Lots of suggestions and methods were proposed for setting up LLs in recreational zones. In different cultures or for different purpose, the characteristics of the LL construction are different, leading to the functional discrepancies among different LLs. In order to setting up effective LLs for specific operational purposes based on their functional outcomes, discerning and evaluating the LL functions are of significance. However, relevant research is currently insufficient. The present study devotes to this field.

## 2.3 Functions and construction of creative linguistic landscapes

**(1) Functions of creative linguistic landscapes.**   Effective LLs can improve the aesthetic quality of recreational zones [31]. Creative design is an important method for producing effective and appealing LLs. Some common LLs cannot arouse the interest of visitors [46], convey emotional messages, or construct scenarios; however, CLLs can.

The CLL contributes to accomplishing a great deal of meaning-making in recreational zones [5]. Firstly, CLLs help to improve spatial expressions of recreational zones. They can enhance the place aesthetics [55], present the local culture [56], form a "creative atmosphere" (surroundings that make visitors experience creativity) [10]; for example, The language used in a CLL can create an exotic, nostalgic, or welcoming atmosphere, among others [42]. Some CLLs can be used to create particular surroundings, for example, CLLs with lively Chinese handwriting in certain recreational zone contribute to constructing a Chinese cultural scenario and a vigorous place [57]. Secondly, CLLs have been constructed in some recreational zones to elicit visitors' feelings and bring them deeper experiences; moreover, creating a CLL is inexpensive relative to its effects [37]. When tourists are unimaginative, boring and ignorant, the CLL can conduce to promoting their tourism experiences [3]. The animated language in the CLL helps to communicate emotions, invoke tourists' cultural memories, impart knowledge, reveal life meaning [34], foster visitors' landscape imagination and wish expression [58], inspire their spiritual identity [56], induce visitors' feelings of warmth, joy, or serenity [59], among others. Interactive elements of CLLs can induce tourists' more mindfulness [49], and promote visitors' interaction with the site [51]. Some CLLs can lead tourists to understand the connotation of language based on the context [41]. Especially for first-time visitors, a distinctive CLL can offer a fresh and authentic experience, and pique their curiosity [42]. Thirdly, the

CLL can play better effect in stimulating and regulating tourists' behavior. With effective discourse, the CLL can promote tourist consumption in the destination [1]; for example, the hand-painted signage representing drug lists make tourists buy pharmaceuticals as souvenir, prioritizing recreation over healthcare [20]. Interesting written language can encourage some individuals' interests to be tourism volunteers [7]. CLLs can also be relevant to safety; for instance, in hazardous water areas, a novel, uncluttered language landscape could be significantly useful in reducing water entry [37].

A variety of CLLs exist, with different functional advantages. Classifying the functions, evaluating and identifying the functional advantages of different types of CLLs can help operators in choosing and creating a corresponding landscape. However, it is difficult to find existing studies that do so. Therefore, this study discusses this topic in detail.

**(2) Construction of creative linguistic landscapes.**   People generally have a great desire for novelty, and CLLs partially satisfy this desire [33]. Additionally, because the LL in some recreational zones doesn't sufficiently reveal the attributes of interactivity, playfulness and emotional experience of the space [25], it needs to be improved through innovation. Moreover, the language used in recreational zones is likely in the permanent process of adaptation to tourists' expectations and interest [6], this also requires the continuous creativities for the LL. Therefore, managers at some recreational sites have begun renewing existing LLs creatively [47]. For the above reasons, CLLs should be designed and placed to make recreational zones more relaxing, emotionally evocative, aesthetically pleasing, healthy, interesting or ecological [23]. In creating CLLs for recreational zones, elements that can be represented innovatively include indigenous languages [60], the shape of distinct local species [48], famous proverbs [61], images of traditional handicrafts (such as Chinese paper-cutting) [35], local traditional writing [18], and famous calligraphy [27], among others.

Firstly, the language usage is a critical factor. In setting up the CLL in recreational zones, emotional and user-friendly language has become a trend [6]; poetic language can be applied to create a romantic atmosphere [1]; familiar idiom can facilitate collective identity, being beneficial for the retention of the place sense [7]. In addition, since the linguistic diversity is being erased in the globalization process, the minority language can be used as a distinctive feature of the place in setting the CLL, and to create place authenticity [41]. Moreover, some film language is conductive to shape the place image in tourists' mind because their emotional attachment has already been built by the movie plot [32]. Secondly, local culture is an effective resource for designing a CLL, and the landscape in tune with the local culture can convey the local style [42], characterize sightseeing routes, and bring historical culture to life [45]. As important cultural elements, local traditional arts are valued in creating a CLL, and local human stories and the natural history can also be reflected in CLLs [46]. Especially in those recreational zones that take outsiders as the main tourist source, where visitors want to learn more about local culture, cultural elements can be presented through indigenous-style CLLs [61]. Thirdly, private LLs have been shown to be more diverse and creative than official ones and also tend to be more fashionable [42]. Official LLs tend to present uniform and rigorous content; however, the language of private LLs is freer and has more interaction functions, tending to show the owner's personality, thought, or specialty, which have the potential to reflect some distinct features or originality and arouse some visitors' interest in exploring [62]. Therefore, private owners' role in facilitating LL innovation should also be considered. Fourthly, researchers also mentioned other elements in setting up CLLs. Sometime, a high-quality, intuitive, and provocative photograph can be presented in a CLL to promote affective communication and amuse the reader [63]; in addition, graphic designs of CLLs are more effective in grabbing visitor attention [51]; furthermore, different colors can be used appropriately to offer visitors a refreshing visual prospect [31] and stimulate changes in their emotions

[64]; and different words carriers (such as stone, wooden board, glass, paper, among others) can be used to bring different senses to tourists, for example, words written inside sand bottles convey a sense that tone in with the ancient city [65].

Functions and content innovation of CLLs are well-regarded in related research; words are the primary content and the most important functional element of CLLs, and different words produce diverse functions; however, research focusing on functional comparison of CLLs with various novel words is scarce, while which can provide essential basis for CLLs configuration in recreational zones, and for people's understanding of the roles of CLLs. To address this gap, the current study systematically analyzes the functions of CLLs with novel words and compares the advantages of those functions.

## 3. Research design

### 3.1 Theoretical framework

Based on relevant research, The words can be divided into two categories: concrete and abstract words [66], representing concrete concepts and abstract concepts respectively. The concrete concepts yield more sensory, material, temporal and spatial expressions, and concrete actions; the abstract concept yield more associations and inner processes (e.g., emotions, introspections, beliefs), relating with empathic concern, and leading to more interaction [67]. By the same token, the language functions in two aspects: it delivers deictic & exact information, or delivers emotional & abstract information [68]. Although the language sign can transmit the above two kind of information concurrently, actually, it always principally transmits the former or latter kind of information. As regards the CLL in recreational zones, it also principally transmits deictic & exact information, or emotional & abstract information. For example, some CLL is for safety tips [37], frequently displaying concrete content; However, some is for creating exotic and nostalgic atmosphere [42], tending to display abstract content. In view of this, this study attempts to discern the functions of CLLs in transmitting "deictic & exact" or "emotional & abstract" information.

Moreover, according to the theory of the three functions of language, expressive function (speaker), appeal function (hearer), representative function (objects) are the basic functions of language [69]. For the CLL, the speaker and hearer mentioned above correspond to the setter and reader of the CLL respectively. Involving some objects, the CLL setters accomplish discursive expression, and this will affect readers' emotion, mood, or activities, among others. Therefore, for evaluating the CLL functions, this study explores whether the desired expression can be accomplished by CLL setters, and whether the reader can be effectively impressed in some aspect. Additionally, in the functional implementation of CLLs, both efficacy and restriction factors exist. For example, a clear theme, interest, readability are efficacy factors, and confliction with the environment, improper placement site, and discrepancy with tourists' demand belong to restriction factors [70]. So, this study discerns these related efficacy and restriction factors, and evaluates the leisure functions of different CLLs accordingly.

Based on the above analysis, this study constructs a theoretical framework (Fig 1) described as follows, on which the inquiry and analysis are conducted. Under the object condition of specific recreational zone, the setter can efficiently express the leisure elements with CLLs; In setting up CLLs, some efficacy and restriction factors will influence the implementation of the CLL functions; Through transmitting "deictic & exact" or "emotional & abstract" information, the installed CLLs appeal to the readers among tourists; Consequently, CLLs contribute to improving spatial expression, and promoting the readers' tourism experiences and behaviors.

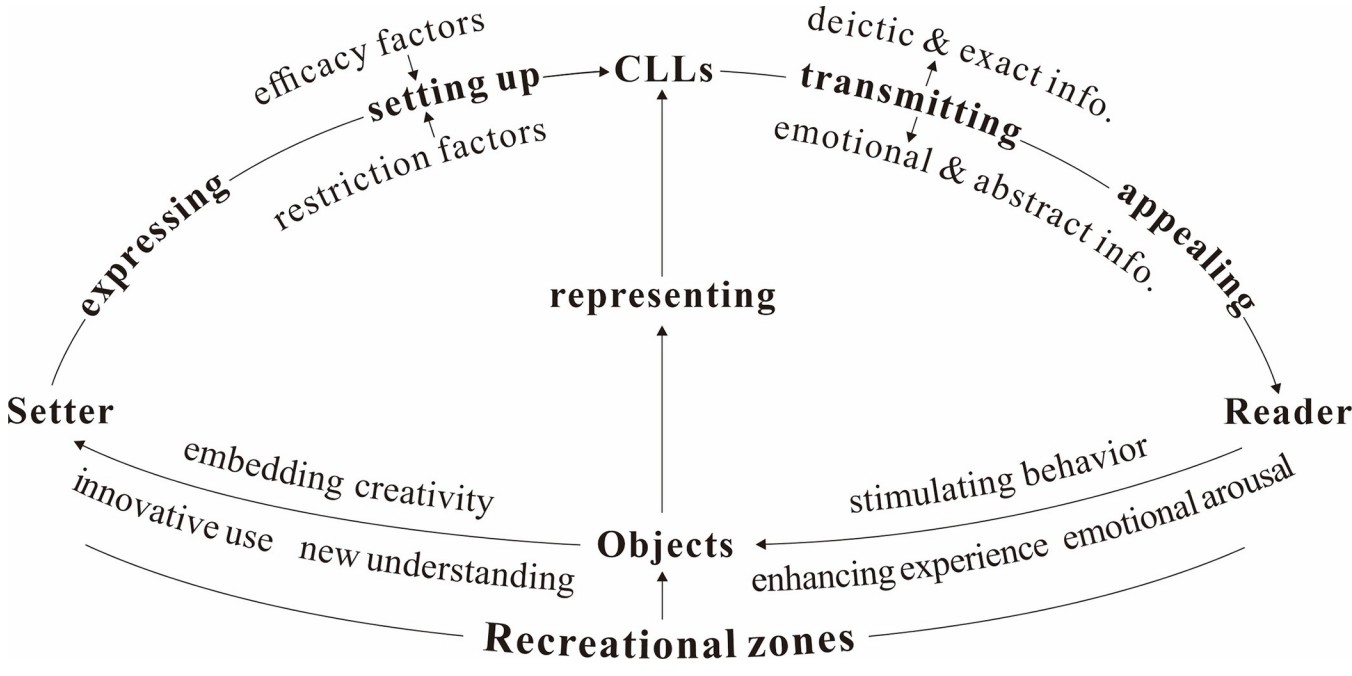

**Fig 1. Theoretical framework.**

## 3.2 Summarizing CLL function types

The authors found numerous images of CLLs with informative content from websites such as Xiaohongshu (www.xiaohongshu.com), Sina Weibo (www.weibo.com), and TikTok (www. douyin.com), which are popular user generated content platforms in China. Some CLLs have a novel appearance, and some show a common appearance but with novel words that reflect their creativity. The functions of CLL are mainly generated by the words used, and the main purpose of this study is to discern the functions of CLLs with different novel words. Presented on a novel or common appearance basis, the same words can attract varying amounts of attention, causing the same words to have different effects. This study selected only CLLs with a common appearance but novel words as its research object, to compare the effects produced by different word language across a consistent CLL appearance type. The authors found and downloaded 364 different satisfactory CLL samples from the above-mentioned websites from April 1–20, 2023. Given that these CLL appearances are highly similar, the functional differences between CLLs are therefore based primarily on the word language. To discuss the functions that these CLLs perform through their words, we formed a discussion group consisting of three authors and three designers from a tourism planning company; the fourth author was responsible for taking notes in the discussion.

As mentioned above, the CLL always principally transmits "deictic & exact" or "emotional & abstract" information [68]. Moreover, through the review on relevant researches noted above, the CLL can mainly play the following 3 kinds of functions in recreational zones: improving spatial expression through effectively representing the site; enhancing tourists' experience through affecting their senses; and stimulating tourists' behavior through effectively guiding and regulating. For these reasons, for each CLL, firstly, the group achieved consensus on which kind of information being conveyed by it; secondly, referring to the above 3 kinds of CLL functions, the group discerned its main specific and particular function through full discussion. Then, the main types of CLL were identified through induction analysis according to

the functional differences between CLLs. What's more, when conducting induction analysis based on pictures or text, qualitative data must be obtained multiple times to achieve saturation of the data [71]. The authors therefore downloaded 100 other relevant pictures from the above-mentioned websites to conduct a saturation test on the pictures used earlier, to see if the obtained pictures could cover the primary CLL function types. If they were not saturated, additional supplementary pictures were found and used.

### 3.3 Summarizing CLL function indicators

As mentioned above, efficacy factors and restriction factors both influence the implementation of CLL functions [70]. What specific factors work on earth? We haven't found the answer from existed researches yet. Given that numerous public comments can fully reflect functional outcomes of CLLs, which always consist of positive and negative evaluations; and kinds of efficacy/restriction factors in CLL function implementation are widely mentioned in plenty of public positive/negative comments. Meanwhile, the related research suggests that extracting themes from adequate qualitative materials is an important method to select evaluation indicators, which can reduce the potential bias in selecting indicators [72]. Numerous public reviews are important qualitative materials, offered by different individuals in different times, being relatively objective. Therefore, this study extracted positive (efficacy) / negative (restriction) indicators from public positive / negative comments on the CLL functional outcomes. The researchers found 102 commentary articles related to the word language of CLL, on Sohu (www.sohu.com), Xiaohongshu (www.xiaohongshu.com), Meipian (www.meipian.cn), and other websites, including related review content, and extracted the content of relevant reviews to obtain a total of 83,578 words of textual data. The researchers first reserved a small amount of data (15,545 words), and then used the qualitative analysis software NVivo 12 to perform coding analysis on another part of the data (68,033 words) to summarize the positive and negative function indicators of CLLs separately from the positive and negative evaluations in the commentators' reviews. The researchers then used the reserved data to test the saturation of the coded material; if the coded material did not pass the saturation test, the authors continued to find and extract relevant material for supplementation.

### 3.4 Obtaining evaluation data for function indicators of various CLLs

Reviews can only show the potential positive and negative functional outcomes of a CLL; they cannot reflect the degree of influence that a specific CLL will have on a particular aspect. Therefore, we also adopted a ranking method to measure which type of CLL had the greatest impact on a particular aspect (such as guiding visitors' emotions or enhancing their cognition). The specific steps were as follows: (1) for each type of CLL, the authors selected three representative images and presented them on a 30×10 cm color photograph; (2) participants were asked to rank the corresponding functional impact of each type of CLL for every function indicator; researchers worked one-on-one with each participant to complete the ranking; (3) for a given function indicator, the type that ranked first/last among $m$ types of CLL, the function evaluation value of the type of CLL was the highest ($m$)/lowest (1); based on this, this study converted the corresponding ranking result into a function evaluation value; (4) for each function indicator for each type of CLL, the evaluation values of all ranking participants were averaged to obtain the final evaluation value; (5) the survey had gotten approval from the Ethics Committee of Yangzhou University Medical College (Approval No. YXYLL-2023-122), and all procedures of the respondent participation were complied with the standards of the Committee; before each voluntary participant began to fill out the questionnaire, the written informed consent had been obtained.

In this study, the recruitment period for this study started from 15 May 2023 to 25 July 2023, we selected individuals with a certain judgment ability and sufficient time for ranking, and whose ranking process could be easily guided and controlled, as the rankers. The participation is voluntary and informed consent form and questionnaire together were distributed to each volunteer. The questionnaire is completed only after the participant knows and endorses the informed consent form. Participants are free to withdraw at any time without giving any reasons. After filling out, it is deemed that the information in it can be used for this research. A total of 157 individuals ended up participating in the ranking. These included 24 tourism management master's students, 15 undergraduates and 7 teachers with tourism management majors; 7 master's students and 3 teachers with journalism and communication majors; 11 master's students and 4 teachers with art design majors; 7 undergraduates and 4 teachers with marketing majors; and 4 teachers with environmental design majors, all from Yangzhou University; 14 tourist guides in Yangzhou were also invited to taking part in the ranking; in addition 57 hotel guests were randomly selected by the researchers from the Songcheng Hotel, the Guesthouse Hotel, and the Meiju Hotel, also in Yangzhou. The ranking values of these 11 participants were either duplicated, missing, or closely similar over multiple indicators. After excluding invalid ranking results, 146 valid ranking datapoints were obtained.

## 3.5 Comparing functions of various CLL types

Based on the participants' evaluating data (presented in S1 File) for functional indicators of the CLL, the fuzzy set can be constructed; then a fuzzy PROMETHEE method was used to compare the functions of different CLL types. This involved the following steps.

**(1) Calculating the weight of each indicator.** The entropy-weighted method [73] was used to calculate the weight of each indicator, which is believed to be capable of addressing mutual impact between indicators.

Firstly, the entropy-weighted method needs based on an evaluation matrix. To this end, this study started with constructing the original evaluation matrix $A$ ($i = 1, 2, \cdots, m$; $j = 1, 2, \cdots, n$), shown in Formula (1), in which, $a_{ij}$ was the composite value of the $j$-th indicator for the $i$-th type of CLL. Then, the normalized matrix $B$ was obtained through normalizing the positive / negative indicator values ($a_{ij}$) by Formulas (2) / (3), shown in Formula (4).

$$A = (a_{ij})_{m \times n} = \begin{bmatrix} a_{11} & a_{12} & \cdots & a_{1n} \\ a_{21} & a_{22} & \cdots & a_{2n} \\ \vdots & \vdots & \vdots & \vdots \\ a_{m1} & a_{m2} & \cdots & a_{mn} \end{bmatrix}, \tag{1}$$

$$b_{ij} = \frac{a_{ij} - \min(a_j)}{\max(a_j) - \min(a_j)}, \tag{2}$$

$$b_{ij} = \frac{\max(a_j) - a_{ij}}{\max(a_j) - \min(a_j)}, \tag{3}$$

$$B = (b_{ij})_{m \times n} = \begin{bmatrix} b_{11} & b_{12} & \cdots & b_{1n} \\ b_{21} & b_{22} & \cdots & b_{2n} \\ \vdots & \vdots & \vdots & \vdots \\ b_{m1} & b_{m2} & \cdots & b_{mn} \end{bmatrix}, \tag{4}$$

In above Formulas (1)–(4), $a_{ij}$ is the average evaluation values of different ranking participants for the $j$-th function indicator of the $i$-th type of CLL; max $(a_j)$ / min $(a_j)$ is the maximum / minimum value of the $j$-th indicator among various types of CLL; $b_{ij}$ is the normalized evaluation value of the $j$-th indicator of the $i$-th type of CLL.

Secondly, the study calculated the indicator weight by following steps: Formula (5) was used to find that among the $j$-th indicator values for various types of CLL, the ratio of corresponding value of each type of CLL; then, Formula (6) was used to calculated the information entropy of the $j$-th indicator; because a lower information entropy of an indicator implies its larger variation degree and a higher weight, so, Formula (7) was used to work out the indicator weights finally.

$$P_{ij} = {}^{b_{ij}} \Big/ \sum i = 1^m b_{ij}, \tag{5}$$

$$e_j = -\frac{1}{\ln m} P_{ij} \ln P_{ij}, \tag{6}$$

$$W_j = \frac{g_j}{\sum_{j=1}^n g_j} = \frac{(1 - e_j)}{\sum_{j=1}^n (1 - e_j)}. \tag{7}$$

In Formulas (5)–(7), $P_{ij}$ is the ratio of the $j$-th indicator value of the $i$-th type of CLL among the $j$-th indicators values of $m$ types of CLL, while $e_j$, $g_j$, and $W_j$ are the information entropy, information utility value, and weight of the $j$-th indicator, respectively.

**(2) Calculating the functional superiority degree of each type of CLL.** PROMETHEE method [74] was used to compare different CLL functions. This method ascertains the superior order among various options through the pairwise comparison, can better address the information warp during analysis process. The analysis included the following steps.

First, for the $j$-th indicator of the $i$-th type of CLL, its average evaluation value ($a_{ij}$) among the investigation participants reflects their comprehensive estimation. Taking $a_{ij}$ as the basic elements, a fuzzy evaluation matrix was constructed, which was same as the matrix $A$ mentioned above. Then, the deviation between $a_{ij}$ and $a_{kj}$ was gotten by Formula (8), being marked as $d_j(a_i, a_k)$.

$$d_j(a_i, a_k) = a_{ij} - a_{kj} \quad (i, k = 1, 2, \cdots, m) \tag{8}$$

Second, referring to relevant research [75], this study applied a revised linear criterion to discriminate the probability of $a_{ij}$ being superior to $a_{kj}$, which was marked as $P_j(a_i, a_k)$, as

shown in Formula (9).

$$P_j(a_i, a_k) = \begin{cases} 0, \ d_j(a_i, a_k) \leq 0; \\ \dfrac{d_j(a_i, a_k)}{\beta d_j(M_j^+, M_j^-)}, \ 0 < d_j(a_i, a_k) \leq \beta d_j(M_j^+, M_j^-) \\ 1, \ d_j(a_i, a_k) > \beta d_j(M_j^+, M_j^-) \end{cases} \tag{9}$$

In Formula (9), $\beta$ is the preference coefficient for strict superiority, and according to relevant research [75], here it is assigned a value of 0.6; $M_j^+$ is the positive ideal solution for the $j$-th indicator, being the maximum comprehensive value of the $j$-th indicator among various types of CLL; $M_j^-$ is the corresponding negative ideal solution, being the corresponding minimum value; the other symbols are the same as mentioned above.

Third, the superiority index of the $i$-th type of CLL to the $k$-th type of CLL was gotten by Formula (10), being marked as $H(a_i, a_k)$. In Formula (10), $W_j$ is the weight of the $j$-th indicator among all indicators, as mentioned above.

$$H(a_i, a_k) = \sum_{j=1}^{n} W_j P_j(a_i, a_k) \quad i, k = 1, 2, \cdots, m; j = 1, 2, \cdots, n \tag{10}$$

Fourth, the net flow of the $i$-th type of CLL was calculated by Formula (11), being marked as $\phi(a_i)$. If $\phi(a_i) > \phi(a_k)$, the former is superior to the latter, and vice versa. Accordingly, the functional superiority order of various types of CLL was obtained.

$$\phi(a_i) = \phi^+(a_i) - \phi^-(a_i) = \sum_{k=1}^{m} H(a_i, a_k) - \sum_{k=1}^{m} H(a_k, a_i) \tag{11}$$

In Formula (11), $\phi^+(a_i)$ is the outflow of the $i$-th type of CLL, implying the overall functional superiority degree of the $i$-th type of CLL to other types of CLL, and a bigger $\phi^+(a_i)$ means a higher corresponding superiority degree; $\phi^-(a_i)$ is the inflow of the $i$-th type of CLL, implying the overall functional superiority degree of other types of CLL to the $i$-th type of CLL, and a smaller $\phi^-(a_i)$ means a higher superiority degree of the $i$-th type of CLL. The other symbols are the same as mentioned above.

## 4. Results and analysis

### 4.1 Main function types of CLL

First, by comparing the contents of 364 different images, the study categorized CLLs into the following nine groups (separately marked as S1) based on function distinctions (Fig 2). The leisure functions of only 3 types of CLL are mainly generated with deictic & exact information, including the types of S6, S7, S8; however, for other 6 types of CLL, their leisure functions are mainly generated with emotional & abstract information, for example, the type for emotional arousal. This indicates that CLLs mainly deliver emotional and abstract information. From another perspective, S2, S7, S8 mainly function in improving spatial expression; S1 mainly function in enhancing tourists' experiences; and S6 mainly function in stimulating tourists' behavior. This indicates that CLLs can play more functions in enhancing tourists' experiences. Furtherly, to test whether the images used for the previous categorization were saturated, another 100 pictures were gathered and categorized in the same way, and no new categories were identified, indicating saturation.

Fig 3 shows the function-type structure of CLL in the focal recreational zones. Several findings can be seen in the figure, as follows: (1) The functional types of CLL are diverse, and the

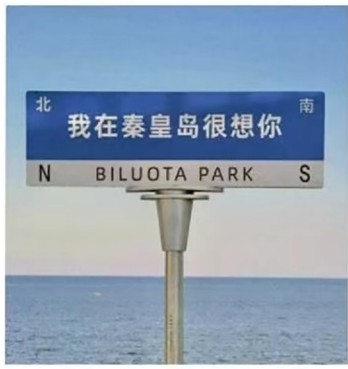
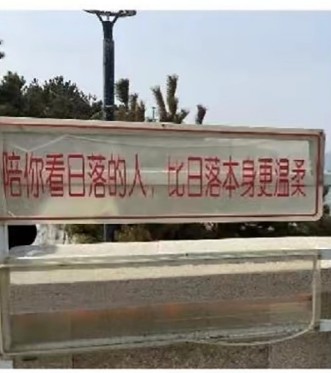
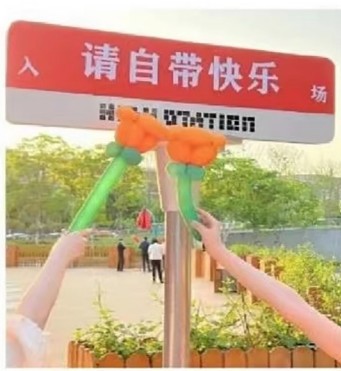

**S1 For emotional arousal**
**Word language:** I miss you so much in Qinhuangdao.
**Site:** Qinhuangdao Biluota Park.
**Source:** http://xhslink.com/xQK6Jp

**S2 For creating atmosphere**
**Word language:** Your companion shows more tenderness than sunset.
**Site:** Yangma Island in Yantai City
**Source:** http://xhslink.com/sBoYKw

**S3 For mood regulation**
**Word language:** Please bring your own pleasure when coming.
**Site:** The Hi Station in Nanjing
**Source:** http://xhslink.com/niC5Us

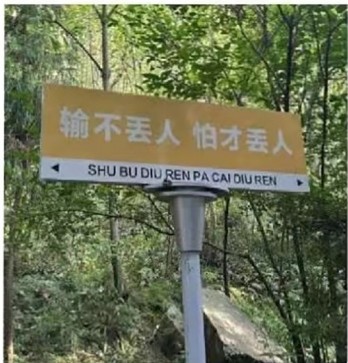
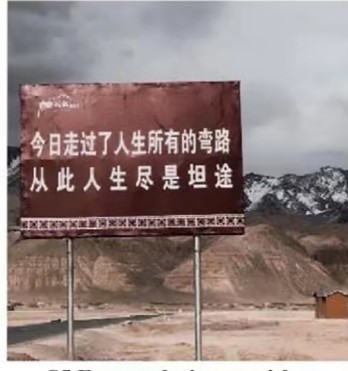
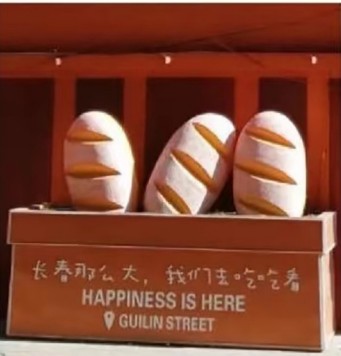

**S4 For mentality guidance**
**Word language:** No shame in losing, only shame in being afraid.
**Site:** Langshan Mountain
**Source:** http://xhslink.com/vjQ0Kw

**S5 For pandering to wishes**
**Word language:** Walk through all det- -ours today and then life is always easy.
**Stie:** Panlong Ancient Road in Kashgar
**Source:** http://xhslink.com/YjpuQw

**S6 For inducing behaviors**
**Word language:** Changchun is so big, let's try the fine food.
**Site:** Guilin Alley in Changchun City
**Source:** http://xhslink.com/8zhyCw

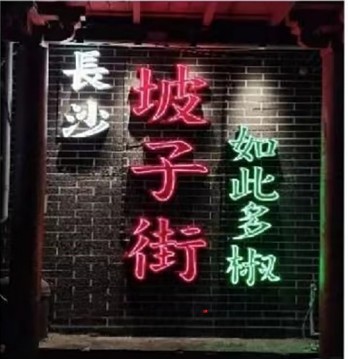
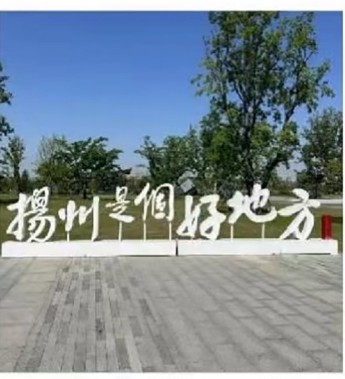
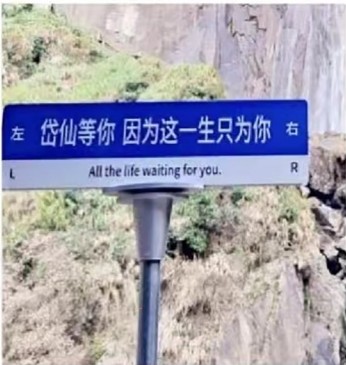

**S7 For labeling site**
**Word language:** Pozi Street with so many chili peppers.
**Site:** Pozi Street in Changsha
**Source:** http://xhslink.com/QjPXet

**S8 For building cognition**
**Word language:** Yangzhou is a nice place to live.
**Site:** Sanwan Park in Yangzhou City
**Source:** http://xhslink.com/DvNfAw

**S9 For resonating with movie lines**
**Word language:** All the life waiting for you—line in 《Chinese Paladin》
**Site:** Fujian Shiniushan Scenic
**Source:** http://xhslink.com/usf6Jp

**Fig 2. The functional types of CLL and the examples.**

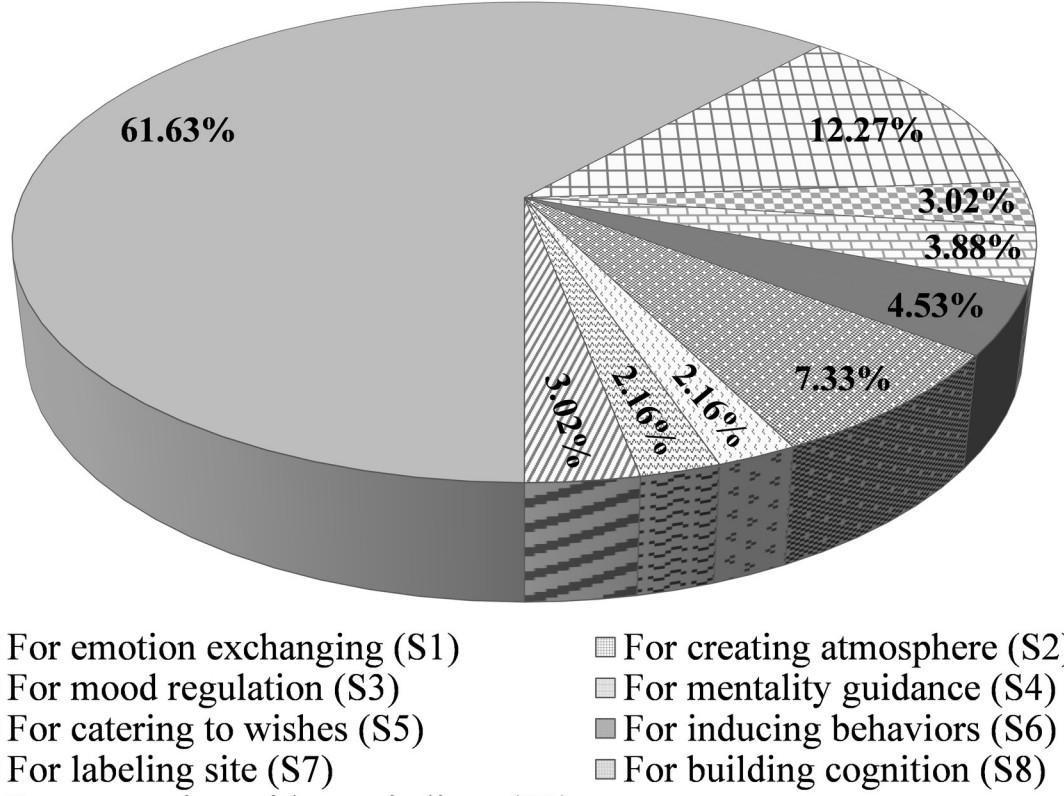

**Fig 3. The proportion of different types of CLL.**

main functions performed by the different types of CLL are obviously distinct. Recreational zones can harness most of the multiple functions of CLLs by adopting diversified types of them. (2) In practice, the applications of different CLL types are imbalanced. S1 comprises the absolute majority and is most prevalent in recreational zones. Owing to the lack of systematic research illuminating the function types of CLL, many operators in recreational zones have not formed a thorough understanding of the function types of CLL, which limits the range of CLLs they can set up. (3) After comparing the relevant images, the study finds that content similarity within S1 is prominent, with "I miss you so much in [XXX place]" and "the wind of missing you blows to [XXX place]" as the main content; these sentences were originally and creatively used by some leisure operators and then were widely copied by other operators, resulting in a monotonous feeling for visitors. This indicates that when setting the CLL in a recreational zone, there are problems of mere imitation and blindness of following trends; that is, the use of personalized creativity is insufficient.

### 4.2 Function indicators of CLLs

And as mentioned above, the CLL probably produce positive or negative effects in recreational zones. Therefore, based on the relevant reviews, this study extracted positive / negative indicators for evaluating CLL functions from public positive / negative comments. We summarized 80 factors that positively affect the functional implementation of CLLs and extracted 22 concepts and 7 categories; additionally, it summarized 30 factors that negatively affect the functional implementation of CLLs, and extracted 10 concepts and 5 categories from them. From

**Table 1. Function indicators of CLLs.**

| Main category (Dimension) | Category (Indicators) | No. | Concept | Coding reference points | Action direction | Weight % |
|---|---|---|---|---|---|---|
| Space construction | The limit of suitable locations | X1 | Not suitable to be placed everywhere at will, not functioning in some places | 126 | Negative | 8.55 |
| | Producing innovative content | X2 | Injecting novel elements, injecting creative elements | 63 | Positive | 8.58 |
| | Manifesting a unique image | X3 | Showing a charming image, forming an image symbol, showing distinction | 59 | Positive | 8.63 |
| | Not easy to update | X4 | Few alternative words, not easy to think of new words | 25 | Negative | 8.38 |
| | Easy to become outdated | X5 | Easy to become outdated | 10 | Negative | 9.62 |
| Recreation experience | Enhancing visitors' experiences | X6 | Creating atmosphere for experience, enhancing spatial vitality, increasing recreational interest, increasing participation experience, deepening tourism memory, forming tourism witness | 118 | Positive | 9.25 |
| | Promoting emotional expression | X7 | Triggering emotions for the other person, triggering emotions for the place, triggering emotions for oneself | 55 | Positive | 7.54 |
| | Obstructing recreational activities | X8 | Occupying space, not allowing other attractions to get enough attention | 34 | Negative | 8.49 |
| | Promoting negative experiences | X9 | Being boring, being awkward, being tasteless | 31 | Negative | 7.47 |
| | Adjusting visitors' mood | X10 | Promoting pleasure, calming the mind, promoting excitement, promoting confidence | 25 | Positive | 7.28 |
| Marketing | Promoting online marketing | X11 | Easy to forward, attracting more attention of readers online | 116 | Positive | 8.45 |
| | Attracting attention on site | X12 | Attracting people to view, attracting people to take photos with it | 67 | Positive | 7.76 |

these 12 categories, 3 main categories were further refined (Table 1). Throughout, running a recreational zone involves 3 actions: space construction, recreation experience, and marketing; and the 3 main categories exactly correspond to these 3 processes (Fig 4). For marketing, only when the target individuals catch sight of the CLL, it can play marketing functions. Nowadays, individuals mainly find the CLL online (in travel notes, propaganda, travel tips, among others) or on site, so, the marketing function of CLLs can also mainly be evaluated from 2 aspects: promoting online marketing, and attracting attention on site. Consequently, the above 12 categories served as functional indicators of CLLs in this study, and the three main categories served as their functional dimensions.

Table 1 shows various indicators influencing the functional outcomes of CLLs. (1) The limits of suitable locations ($X1$, indicating the number of sites where the CLL functions), enhancing visitors' experiences ($X6$), and promoting online marketing ($X11$) are highly recognizable function indicators of CLLs, with the highest mention rates in various reviews. This demonstrates that the public has fully recognized some positive and negative functional outcomes of CLLs. (2) With a medium mention rate in various reviews, producing innovative content ($X2$), manifesting a unique image ($X3$), promoting emotional expression ($X7$), and attracting attention on site ($X12$) are medium-recognition indicators; and with a lower mention rate, easy to update ($X4$), easy to become outdated ($X5$), obstructing recreational activities ($X8$), promoting negative experiences ($X9$), and adjusting visitors' moods ($X10$) are still low-recognition function indicators of CLLs. This demonstrates that many people haven't formed a comprehensive understanding of the CLL functions, and many corresponding functional outcomes have not yet received sufficient attention. In this context, it is necessary to reveal the diversified functions of CLLs. (3) Although the attention given to each functional indicator of the CLL in

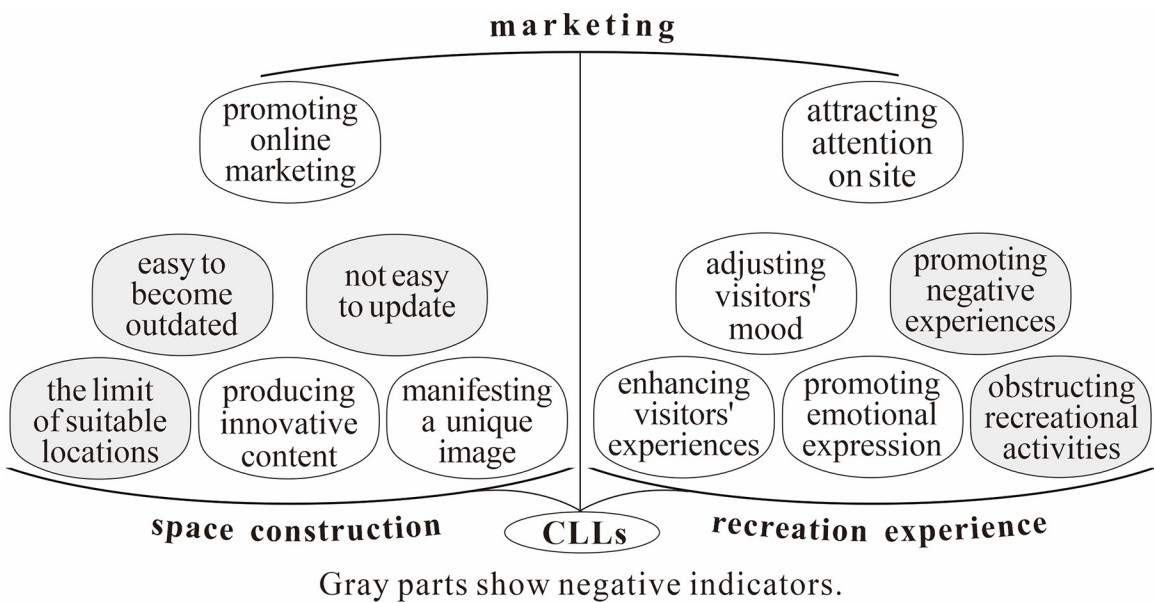

**Fig 4. The indicator structure for evaluating CLL functions.**

practice varies significantly, the results of the indicator weight analysis based on the entropy-weighted method show that the importance of each indicator does not differ significantly in their comprehensive influence on the function implementation of the CLL. For example, although whether the function of a CLL is easy to become outdated ($X5$) has not yet received sufficient attention, it has a significant impact on the function implementation of the CLL, because it is closely tied to whether the CLL can continue to perform its functions.

### 4.3 Evaluation results of the function indicators of different types of CLLs

If the data is lack of explanatory power and validity, effective and convincing conclusions can hardly be drawn. For further analysis, the above ranking results need to meet the following 2 requirements: for specific function indicator, different types of CLL reveal different leisure function outcomes, and the function outcome ranking of various types of CLL made by different participants show some consistency. The Friedman test [76] and Kendall tests [77] can be used to find whether the ranking results meet the above first and second requirement respectively. The results of the Friedman and Kendall tests (using SPSS 26.0) on values ranked by the participants for specific function indicator of each type of CLL are shown in Table 2. In the critical values table for the chi-squared test, the critical value of chi-square was 15.507, at a significance level of 0.05 and a degree of freedom of 8. If the Friedman statistic is more than the critical value with a higher significance level, the test is passed. As shown in Table 2, for each specific function indicator, the chi-square values of the Friedman test on the evaluation values of various types of CLL given by the ranking participants were all >15.507, with an asymptotic significance of <0.001, indicating differences in the relevant functional outcomes of the various types of CLL. In the Kendall test, a concordance coefficient more than 0.6 at a high significance level implies a stronger consistency [77] of the ranking results given by different individuals. Table 2 showed that, for each indicator, the corresponding evaluation values passed the Kendall consistency test, indicating that the ranking participants showed some consistency in ranking the functional outcomes of the different types of CLL. The Friedman and

**Table 2. Results of Friedman and Kendall tests on evaluation values.**

| Function indicators | Friedman test | | Kendall test | | Degree of freedom |
|---|---|---|---|---|---|
| | Chi-square value | Asymptotic significance | Concordance coefficient | Asymptotic significance | |
| X1 | 842.258 | 0.000 | 0.721 | 0.000 | 8 |
| X2 | 932.093 | 0.000 | 0.798 | 0.000 | 8 |
| X3 | 868.538 | 0.000 | 0.744 | 0.000 | 8 |
| X4 | 788.791 | 0.000 | 0.675 | 0.000 | 8 |
| X5 | 820.499 | 0.000 | 0.702 | 0.000 | 8 |
| X6 | 817.092 | 0.000 | 0.700 | 0.000 | 8 |
| X7 | 924.425 | 0.000 | 0.791 | 0.000 | 8 |
| X8 | 883.023 | 0.000 | 0.756 | 0.000 | 8 |
| X9 | 969.812 | 0.000 | 0.830 | 0.000 | 8 |
| X10 | 929.787 | 0.000 | 0.796 | 0.000 | 8 |
| X11 | 874.276 | 0.000 | 0.749 | 0.000 | 8 |
| X12 | 849.624 | 0.000 | 0.727 | 0.000 | 8 |

Kendall tests indicated that the above ranking results were discriminative and consistent, and were capable for further analysis.

Fig 5 shows the comprehensive evaluation results (between 1 and 9) for various functional indicators of the different types of CLL (the negative indicator values have already been reversed in order to ensure the comparability among various indicators, so they need to be understood conversely). The advantageous function indicators of different types of CLL significantly differ from each other, which further suggests that recreational zones need to enrich types of CLL to provide full play to the various functional advantages of CLLs by type. Some CLL types have outstanding positive functional advantages. For example, S2 has outstanding functional advantages for producing innovative content ($X2$) in recreational zones and manifesting their unique images ($X3$), whereas S3 has outstanding functional advantages for enhancing visitors' experience ($X6$) and promoting their emotional expressions ($X7$). At the same time, negative functional outcomes of some CLL types are also prominent. Among others, S1 is very easy to become outdated ($X5$), for example, easy to copy and thus lose novelty

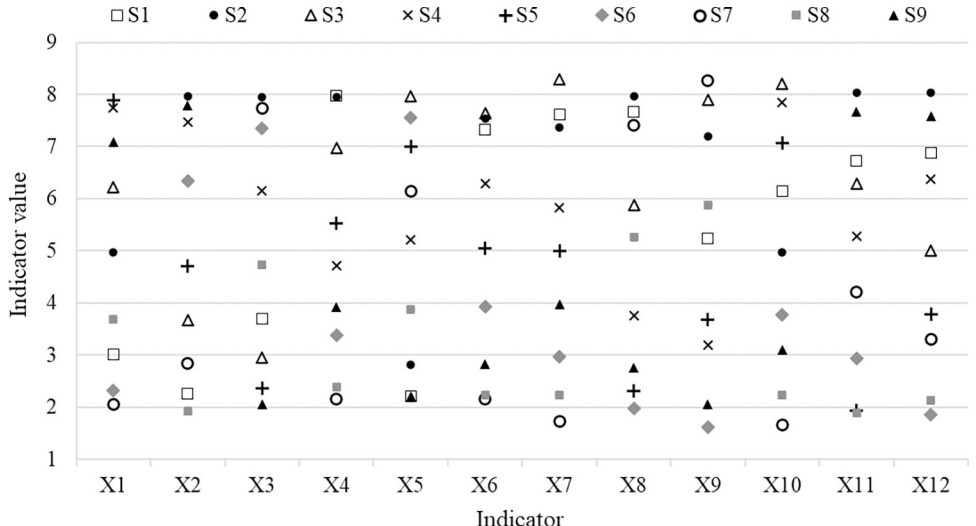

**Fig 5. Function indicator scores of different types of CLL.**

and visitors' attention; it is not easy to update ($X4$) for S6, for example, it is not easy to think of new words for it; and S5 has a high potential to promote a negative experience ($X9$), for example, visitors may feel that the operator is imposing certain intentions on them. When setting up CLLs in a recreational zone, the study results can provide references for consciously embodying the outstanding positive functional outcomes of the CLL while reducing or avoiding its more obvious negative functional performances.

## 4.4 Functional sequences of CLL types

**(1) Comprehensive functional sequence of CLL types.** The ranking results for the comprehensive leisure functions of the nine CLL types are listed in Table 3, which are obtained based on the ranking participants preference matrix for CLL types. First, three types of CLL (S2, S3, and S4) can infect, regulate, and induce people's moods, that is, they primarily have a mood-guiding function, with the best ranking of overall functions among all types of CLL. This indicates that mood guidance is the most prominent function of CLLs, although this function has not been paid enough attention as is shown above. The main orientation when configuring a CLL in a recreational zone thus needs to be the effects on human mood. Second, three types of CLL—S1—play an emotionally guiding role, being in the middle of the comprehensive function-ranking sequence of various types of CLL. This suggests that emotional guidance for visitors is also an important aspect of CLL. When configuring CLLs in a recreational zone, their emotionally guiding functions should also be possibly embodied, to give full play to the positive effects of CLLs. Third, three types of CLL, S7, S6, and S8, play a cognitive guidance role, that is, they can guide visitors to form cognitions concerning places, behavioral ideas, and other aspects, respectively. However, the comprehensive functions of these three types of CLL ranked below those of the other types. This further suggests that the main functional advantage of CLLs is not cognitive guidance for visitors, but guidance for visitors' moods and emotions, being different from common linguistic landscapes. Thus, when setting up CLLs in recreational zones, their cognitive guidance function can be considered, but should not be focused on as the main orientation.

**(2) Functional sequences of CLL types in different dimensions.** Based on the functional sequences of CLL types in different dimensions (Table 4), the following conclusions can be drawn. First, in terms of leisure space construction, the functional advantages of S4 are the most prominent. For this type of CLL, the limit of suitable locations ($X1$) is lower, and its function is not likely to become outdated ($X5$) (Fig 5). This reduces pressure on its placement and updating during constructing and maintaining. Additionally, S2 and S3 also have some functional advantages in terms of leisure space construction. If the main purpose is to facilitate and

**Table 3. The preference matrix and functional sequences of CLL types.**

| CLL type | S1 | S2 | S3 | S4 | S5 | S6 | S7 | S8 | S9 | Outflow ($\phi^+$) | Inflow ($\phi^-$) | Net flow ($\phi$) | Sort results |
|---|---|---|---|---|---|---|---|---|---|---|---|---|---|
| S1 | **0** | 0.027 | 0.133 | 0.309 | 0.469 | 0.628 | 0.486 | 0.551 | 0.487 | 3.090 | 2.092 | 0.999 | 4 |
| S2 | 0.341 | **0** | 0.348 | 0.456 | 0.650 | 0.714 | 0.651 | 0.749 | 0.561 | 4.469 | 0.930 | 3.538 | 1 |
| S3 | 0.318 | 0.220 | **0** | 0.371 | 0.503 | 0.724 | 0.564 | 0.721 | 0.581 | 4.002 | 1.290 | 2.712 | 2 |
| S4 | 0.347 | 0.187 | 0.230 | **0** | 0.382 | 0.541 | 0.562 | 0.688 | 0.450 | 3.388 | 1.710 | 1.678 | 3 |
| S5 | 0.257 | 0.206 | 0.065 | 0.082 | **0** | 0.356 | 0.461 | 0.554 | 0.347 | 2.328 | 2.880 | −0.552 | 5 |
| S6 | 0.268 | 0.096 | 0.150 | 0.094 | 0.164 | **0** | 0.270 | 0.383 | 0.226 | 1.652 | 3.723 | −2.071 | 8 |
| S7 | 0.253 | 0.112 | 0.129 | 0.224 | 0.298 | 0.229 | **0** | 0.331 | 0.342 | 1.917 | 3.520 | −1.604 | 7 |
| S8 | 0.100 | 0.029 | 0.043 | 0.086 | 0.169 | 0.191 | 0.068 | **0** | 0.242 | 0.929 | 4.411 | −3.481 | 9 |
| S9 | 0.208 | 0.052 | 0.192 | 0.088 | 0.246 | 0.341 | 0.458 | 0.434 | **0** | 2.019 | 3.237 | −1.219 | 6 |

**Table 4. The functional sequences of CLL types in different dimensions.**

| Types of CLL | In space construction aspect | | | | In recreation experience aspect | | | | In marketing aspect | | | |
|---|---|---|---|---|---|---|---|---|---|---|---|---|
| | Outflow ($\phi^+$) | Inflow ($\phi^-$) | Net flow ($\phi$) | Sort results | Outflow ($\phi^+$) | Inflow ($\phi^-$) | Net flow ($\phi$) | Sort results | Outflow ($\phi^+$) | Inflow ($\phi^-$) | Net flow ($\phi$) | Sort results |
| S1 | 0.636 | 1.724 | −1.089 | 8 | 1.756 | 0.276 | 1.480 | 3 | 0.699 | 0.091 | 0.608 | 3 |
| S2 | 1.707 | 0.698 | 1.009 | 2 | 1.840 | 0.232 | 1.608 | 2 | 0.922 | 0.000 | 0.922 | 1 |
| S3 | 1.438 | 0.890 | 0.548 | 3 | 2.060 | 0.134 | 1.925 | 1 | 0.505 | 0.266 | 0.238 | 5 |
| S4 | 1.682 | 0.582 | 1.100 | 1 | 1.171 | 0.882 | 0.289 | 4 | 0.534 | 0.245 | 0.289 | 4 |
| S5 | 1.392 | 0.899 | 0.493 | 4 | 0.850 | 1.192 | -0.342 | 5 | 0.086 | 0.789 | −0.703 | 7 |
| S6 | 1.359 | 0.940 | 0.419 | 5 | 0.246 | 1.917 | −1.671 | 9 | 0.047 | 0.866 | −0.819 | 8 |
| S7 | 0.874 | 1.477 | −0.602 | 7 | 0.853 | 1.411 | −0.558 | 6 | 0.189 | 0.633 | −0.443 | 6 |
| S8 | 0.409 | 1.900 | −1.491 | 9 | 0.514 | 1.586 | −1.072 | 7 | 0.006 | 0.924 | −0.918 | 9 |
| S9 | 0.963 | 1.350 | −0.386 | 6 | 0.211 | 1.870 | −1.659 | 8 | 0.845 | 0.018 | 0.827 | 2 |

simplify construction, these types of CLL should be prioritized. Second, in terms of the recreational experience, S3 has the most functional advantages, followed by S1. These types of CLL have more functional outcomes in enhancing visitors' experiences (*X*6), promoting their emotional expressions (*X*7), and adjusting their moods (*X*10) (Fig 5). If a CLL is to be set up in a recreational zone primarily to enhance the recreational experience, priority should be given to S3, while S2 and S1 should also be given attention. Third, in terms of marketing, the functional advantages of S2 are the most prominent, followed by those of S9 and S1. If the purpose of establishing a CLL is primarily to promote leisure marketing, these types of CLL should be prioritized.

## 5 Discussion and conclusion

### 5.1 Discussion

The LL is believed a tool for tourists to go through characteristic experience [41]. Additionally, linguistic creativity promotes efficient communication, so the CLL deserves to be given enough attention [78]. Actually, creative language is always used for making humor, underlining important information, manifesting an attitude, or entertaining others, among others [79]; and currently, the CLL have become an important matter in some recreational zones. However, what is the functions of CLL in recreational zones is still shortage of research. This study makes some exploration in this aspect. Previous research stressed that The LL elevates language as the resource [8], and this further research contributes to leading some operators to sufficiently use effective language as important resource in recreational zones. The discussion mainly focuses on the following points.

**(1) The function types of CLL.** Language is the medium par excellence for representation and communication [34], and CLLs can naturally convey lots kinds of meaning, and their function are various [5]. Accordingly, this study clarifies the main function types of CLLs existing in recreational zones and their functional distinctions. This is a theoretical summary for current CLL setting, and can also contribute to the CLL installation practice mainly in two aspects. First, although there are many different CLL types, the distribution of CLL types in practice is severely imbalanced, in that S1 constitutes the absolute majority. This study summarizes the function type structure of CLLs, can help relevant personnel recognize the multiple function types of CLLs, so as to promote the rational configuration of their types in recreational zones and take advantages of various types of CLLs in different ways. Second, CLLs involve multiple function types [36], which also confuses operators concerned with

choosing and matching the corresponding CLL types. This may lead the CLL configuration to fail to satisfy the needs of visitors [38]. In this study, comparative analysis of the functions of different types of CLL helps to eliminate this confusion.

**(2) The function outcomes of CLLs.** The primary roles of conventional linguistic landscapes in recreational zones are to provide information [11], direct cognition, and regulate behavior [22]. Nevertheless, responding to the CLL configuration practice, the present study discusses the CLL features in functional outcomes, which contributes to eliciting adaptive goal in setting up the CLLs.

Early research claimed that language functions correspond to human's inner experience, meaningful behavior, and culture [69]. Following this, the study discovers the functional characteristics of the CLL: the functions of most types of CLL are mainly related with the inner experience, and the main functional advantages of CLLs are mood and emotion guidance. Precisely for this reason, some operators in recreational zones who try to guide visitors' moods and emotions attach great importance to the function of the CLL [58]. Previous research found that specific atmosphere could be efficiently created by CLLs [5], including historical authentic atmosphere [32], socio-spatial authentic atmosphere [41], romantic atmosphere [1], exoticism [26], among others. Similarly, the present study concludes that the CLL for creating atmosphere had the outstanding functional outcomes. Additionally, relevant research found that the CLL are always playfully pragmatic [80], and this study confirmed this again; the CLL plays important functions in an entertaining way.

Unexpectedly, the tourism marketing function of LL is not often mentioned in relevant researches; however, this study found a prominent marketing function of the CLLs; perhaps this is a significant functional difference of the CLL from the LL. Furthermore, previous research proposed that tourism operators need to use the CLL to increase tourist consumption [1]; however, this study finds that the CLLs don't directly show prominent functions in this aspect.

**(3) The function optimization in setting up CLLs.** Due to the lack of relevant knowledge, the operators show some randomness in setting up CLLs [43]. Considering this, the study contributes to providing knowledge for operators concerned. According to this study, some points should be emphasized to optimize the CLL configuration.

The present study insists that when setting up the CLLs in recreational zones, their emotional functions need to be emphasized; this is also an echo for the existed research, which enunciates that using emotional and friendly language is becoming a trend in tourist sites [6]. Related research emphasized the utilization of interactive elements in setting up the LLs [49], and linguistic creativity is more likely to occur in an interactive situation [79]; correspondingly, according to this study, for setting up the CLLs, the interactive elements are particularly important in guiding tourists' emotion and adjusting their mood. The creativity depends on the functional requirement [79]; consistent with this, this study advocates different construction orientation after different intention for the CLLs. Additionally, this study also gives extra proposals in setting up CLLs, for example, firstly focusing on the functional advantages and then balance the function types, choosing the CLL types based on the functional dimensions of space construction, recreational experience, or marketing, to meet the operation intention. These are conductive to setting up the CLLs in recreational zones.

Moreover, this study contributes to address some practical problems in setting up the CLLs. First, some research found that in many recreational zones, the LLs didn't play enough function in promoting interaction, playfulness and emotional experience [25], and are lack of linguistic aesthetic [44]; this study indicates that setting up CLLs are helpful for addressing this. Second, related research believed that an avalanche of language signs may brought about confusion for tourists [3]; this study identifies the functional advantages of CLLs, can increase the

operator's awareness that which functions should be focused on in setting up CLLs in specific recreational zone, contributes to dealing with the above problem. Third, creativity can generate many benefits [81], but may also have some negative effects, for example, related research mentioned visual pollution caused by creative language signage [45]; Constructively, this study presents a comprehensive overview of the positive and negative functional indicators of CLLs, and developed a framework for functional evaluation; this contributes to helping relevant people gain a deeper and more comprehensive understanding of CLL functions, and guiding them to optimize the CLL setting.

## 5.2 Management implications

According to the findings of this study, when setting up the CLL in a recreational zone, the objective should be to coordinate and optimize the functions of various creative language signs, and the mode of focusing and coordinating the functions of CLLs (Fig 6) should be carried out.

**(1) Function focuses in setting the CLL.**   The results of this study show that mood and emotion guidance are two of the most advantageous leisure functions of CLLs in recreational zones. Focusing on these two kinds of functions can give full play to the corresponding functional advantages when setting up the CLL. According to the functional sequences of different types of CLL, mood guidance should be regarded as the primary functional focus target, and the salient mood guidance functions of S2, S3, and S4 for visitors should be fully exploited. Emotion guidance should be regarded as the second functional focus target in setting the CLL, and the salient emotion guidance functions for visitors of S1, S5, and S9 should be implemented. That is, when setting the CLL, the operators should firstly/secondly target the implementation of the mood/emotion guidance functions. On the premise of function focusing, an appropriate amount of creative language signs with the advantages of cognitive guidance, such as S6, S7, and S8, should also be configured to balance the core functions and other diversified functions of CLLs as much as possible.

**(2) Function coordination in setting the CLL.**   When setting up CLLs in a recreational zone, with the goal of maximizing the leisure function, the superior functions should be focused on, while, the functions of different types of CLL should also be coordinated, as well as the functions in different aspects for certain type of CLL.

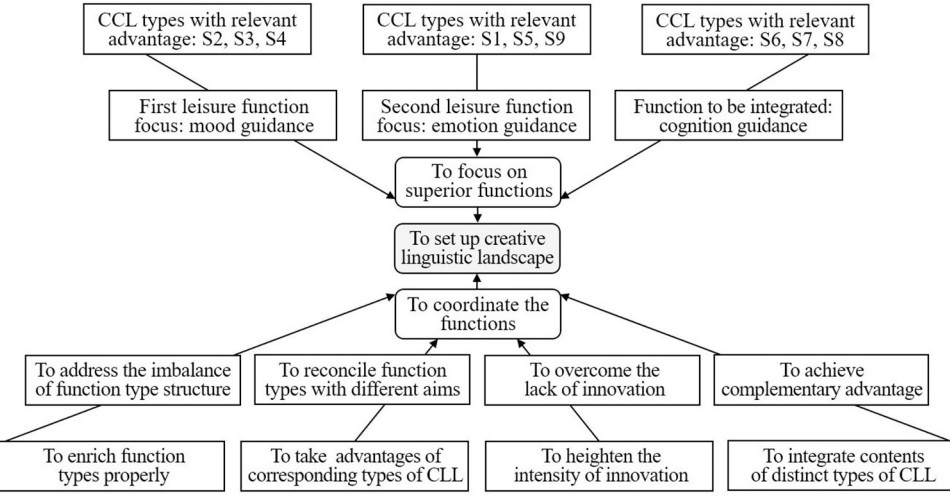

**Fig 6. Mode of focusing and coordinating functions of CLLs.**

Currently, there are nine common types of CLL with different functions, all of which contribute to the improvement of leisure functions in recreational zones. However, the CLL type structure is seriously imbalanced, dominated by S1 only. This makes the leisure functions of the corresponding CLL system less comprehensive and insufficient. When setting up CLLs in a recreational zone, with the aim of achieving a better type structure, diversified function types of CLL should be exploited appropriately to make the functions richer and more diverse, and to optimize the overall function of the CLL system. For example, in practice, S2, S3, and S4 only occupy a small proportion of CLLs, but they have more prominent functions. Therefore, their shares need to be enlarged. In addition, the CLL function types need to be coordinated with the targeted goal clearly defined by the operators. For example, serving simplifying construction and improving visitors' experiences can be targeted respectively by prioritizing S4 and S3, in the CLL configuration.

As mentioned above, the content similarity of many creative language signs is very prominent, showing a lack of individualized creativity, and the phenomenon of simply imitating and blindly following is a serious issue. This results in fewer functional distinctions among different language signs, and weakens the functional outcomes of CLLs. Therefore, when setting up a CLL in a recreational zone, individualized creativity should be emphasized as a tool for improving the functional coordination of CLLs. In addition, as shown in Fig 5, some positive/negative functional performances of some CLL types are weaker/outstanding; for example, S1 has a weaker function of producing more innovative content, and has an outstanding potential to become outdated in a recreational zone. Therefore, when setting up a CLL, negative function reduction and positive function promotion should also be regarded as functional coordination orientations to strengthen the overall function of the CLL. Setting composite language content is likely to be helpful, for example, adding content for labeling sites (corresponding to S7) to S1 (for emotional arousal) can partly reduce the potential of S1 to become outdated; and adding content for creating atmosphere (corresponding to S2) to S7 can partly enhance the potential of S7 to generate some novelty.

## 5.3 Conclusion

Creative linguistic landscapes are emerging in many recreational zones as a stimulus to leisure, but studies on their leisure function are few. Therefore, this study was conducted. Nine leisure function types of CLL in practice were summarized, the functional evaluation indicators system was constructed that consisted of seven positive and five negative functional indicators, and the leisure function outcomes of different types of CLL were compared. This study found that currently the function type structure of CLLs urgently needed to be balanced, their positive functional outcomes needed to be enhanced, and the negative functional performances weakened or eliminated significantly, and its advantageous functional outcomes—guiding visitors' mood and emotion—should be focused on as the important leisure elements in recreational zones. In setting CLLs, the mode of function focusing and coordinating is effective for achieving the above-mentioned objectives. This study is helpful to deepen people's cognitions around CLL, to provide references for the optimization of CLL configuration in recreational zones, and to promote further research on related matters.

## 5.4 Limitations and future research

Research on CLLs is still in its early stages [40], and many issues need to be further explored. For instance, new digital language signage is appearing in recreational zones [82], and its functional impacts urgently need to be revealed. Additionally, there are also CLLs with a novel appearance in some important recreational zones (such as the Gornergrat of Switzerland, the

Radiator Springs of America, the Mt. Yuntai of China, and so on); the significant difference in the leisure function outcomes between creative language signs with novel and common appearances also needs to be further discussed. Moreover, additional empirical investigations are required to test the true effects of setting up CLLs in recreational zones [47]. This should be the focus of future studies.

## Supporting information

**S1 File. Research data.**
(XLSX)

**S1 Questionnaire.**
(DOCX)

## Acknowledgments

The authors are thankful to the functions identifying and ranking participants for creative linguistic landscape.

## Author Contributions

**Conceptualization:** Kun Sun, Xiaoli Tian, Bing Hou.

**Formal analysis:** Xiaoli Tian, Jing Xia.

**Funding acquisition:** Bing Hou.

**Investigation:** Xiaoli Tian, Jing Xia, Qing Li.

**Methodology:** Kun Sun.

**Project administration:** Bing Hou.

**Supervision:** Kun Sun, Bing Hou.

**Validation:** Kun Sun, Xiaoli Tian.

**Visualization:** Jing Xia.

**Writing – original draft:** Kun Sun.

**Writing – review & editing:** Kun Sun, Xiaoli Tian, Qing Li.

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
