## [Decision Letter · Decision Letter 0]

16 Oct 2023

PONE-D-23-26514Promoting leisure functions through setting creative linguistic landscapes in recreational zonesPLOS ONE

Dear Dr. Tian,

Thank you for submitting your manuscript to PLOS ONE. After careful consideration, we feel that it has merit but does not fully meet PLOS ONE’s publication criteria as it currently stands. Therefore, we invite you to submit a revised version of the manuscript that addresses the points raised during the review process.

We look forward to receiving your revised manuscript.

Kind regards,

Youssef El Archi, PhD.

Academic Editor

PLOS ONE

Journal Requirements:

Reviewers' comments:

Reviewer's Responses to Questions

**Comments to the Author**

1. Is the manuscript technically sound, and do the data support the conclusions?

Reviewer #1: Yes

Reviewer #2: Yes

Reviewer #3: Partly

2. Has the statistical analysis been performed appropriately and rigorously? 

Reviewer #1: Yes

Reviewer #2: Yes

Reviewer #3: N/A

3. Have the authors made all data underlying the findings in their manuscript fully available?

Reviewer #1: Yes

Reviewer #2: Yes

Reviewer #3: No

4. Is the manuscript presented in an intelligible fashion and written in standard English?

Reviewer #1: Yes

Reviewer #2: Yes

Reviewer #3: No

5. Review Comments to the Author

Reviewer #1: Thank you for the opportunity to review this excellent paper. The article is well-written and structured. The results and conclusions are useful. I recommend to publish. However, 49 citations are not enough. More references are needed, and please add a more detailed literature review.

What about the general linguistic situation China? Please compare with the EU and worldwide.

E.g.

https://www.sciencedirect.com/science/article/pii/S1877042812052408

and

https://dspace.kmf.uz.ua/jspui/handle/123456789/1514

Etc.

Reviewer #2: Creative Linguistic Landscapes (CLLs) are increasingly prevalent in recreational areas, but determining how to establish them has become a significant challenge that hinders the scientific planning of such areas. This study utilized image data and online reviews to identify leisure function categories and functional evaluation indicators for CLLs. The results of the leisure function of CLLs were ranked using the entropy weighted TOPSIS method. Based on this, an innovative CLL configuration optimization model was proposed. However, there are still some areas for improvement in this study, including:

(1) While this study has successfully summarized the function indicators of CLLs, there is a need to further clarify the screening basis and classification criteria of these indicators to enhance the scientific rigor of the paper. For instance, the marketing function category, which currently only includes promoting online marketing and attracting attention on-site, may not be comprehensive enough.

(2) Section 5.1 of the paper lacks depth and fails to highlight the main focus of the discussion. To improve the quality of the paper, it is recommended to emphasize the differences between this study and previous research and to clearly articulate the innovative contributions of this study.

(3) The authors should carefully read over their manuscript. Spelling and grammar errors should be corrected. The review of a native speaker or language editing services would be helpful.

Reviewer #3: Problem formulation is quite vague. Function distinctions of the CCL must be presented explicitly. It is not clear enough why we have these nine groups? Not all links work properly shown in the table 1. It should be useful to present the images of all types of the CCL. The information concerning function indicators must be structured showing the reasoning of the chosen characteristics from the theoretical and implementational views of the point. All these parts of the text must be moved to theoretical part of the manuscript.

How the weights presented in table 2 are calculated?

The need and the analysis of the results in the table 3 must be clarified.

What metrics are applied to evaluate the considered cases of the CCL?

How the vagueness of the initial information has been taken inti account?

Nowadays the different fuzzy approaches have been proposed to consider the uncertainty of the initial information. Therefore, there is no any novelty concerning the solution method of this decision making problem.

6. PLOS authors have the option to publish the peer review history of their article (what does this mean?). If published, this will include your full peer review and any attached files.

Reviewer #1: No

Reviewer #2: No

Reviewer #3: No

---

## [Author Response · Author response to Decision Letter 0]

7 Dec 2023

Response to first peer review feedback

Dear expert:

Thank you very much for your approval and valuable advices, which are very helpful for our improving the paper, as well as the important guiding significance to our researchers. We have carefully read and studied the comments, and accordingly have made correction which we hope meet with approval. To facilitate your review, we highlighted the primary revisions in track changes mode, and in the response, we also noted down the line numbers where the revision had been made. The main revisions are as follows. We kindly request your review.

Reviewer Comment 1: 49 citations are not enough. More references are needed, and please add a more detailed literature review.

Response: thanks for your comments, which makes us notice the necessary to further enrich and improve the literature review. Accordingly, we have added more than 40 citations in this revision. Based on these references, we enlarged the “introduction” part from 405 word to 744 words (the corresponding contents lies in lines from 26 to 118), and enlarged the “literature review” part from 1271 words to 2207 words. We found that after the revision, the literature review became more detailed (the corresponding contents lies in lines from 128 to 435).

Reviewer Comment 2: What about the general linguistic situation China? Please compare with the EU and worldwide. 

Response: thanks for your comments and providing the clues for relevant information. We have made the revision accordingly (the corresponding contents lies in lines from 269 to 293).

The main relevant content after revision is as follows. The first characteristic of the linguistic landscape in China reflects in the aspects of language types: “Comparing with English-speaking countries, in China, more types of language are used in recreational zones; especially, the language used on bottom-top signage is more diverse (Zhang, 2020); meanwhile, the font types with obvious distinction are very rich.” The second characteristic of the linguistic landscape in China reflects in the strong subjectivity in choosing the language type: “the operators show strong subjectivity in choosing the language type, for example, some use Korean in the signage, but others use German or Malay, among others (Shi, 2021).” The third characteristic of the linguistic landscape in China is the following: because of the feature of Chinese characters, “the effect of Chinese signage is always improved through enhancing its capacity to express the emotion, symbolic meaning, imagery and spirit”, this shows some difference from English signage, because “the English letters can be changed more flexibly in shape and size, so, the effect of English signage can be improved through the interesting and artistical design of the letter shape and size (Zhu, 2014).” These characteristics also makes the linguistic landscape in China have some obvious distinctions.

 

Response to second peer review feedback

Dear expert:

Thank you very much for your validation and valuable comments. Your comments are all valuable and very helpful for our revising and improving the paper, as well as the important guiding significance to our researchers. We have studied your comments carefully and accordingly have made correction which we hope meet with approval. To facilitate your review, we highlighted the primary revisions in track changes mode, and in the response, we also noted down the line numbers where the revision had been made. The main revisions are as follows. We kindly request your review.

Reviewer Comment 1: While this study has successfully summarized the function indicators of CLLs, there is a need to further clarify the screening basis and classification criteria of these indicators to enhance the scientific rigor of the paper. For instance, the marketing function category, which currently only includes promoting online marketing and attracting attention on-site, may not be comprehensive enough.

Response: thanks for your comments. According to your comments, we made corresponding revisions. We request your review.

First, we further clarified the screening basis and classification criteria of the indicators of CLLs. 

(1) efficacy factors and restriction factors both influence the implementation of CLL functions (Wang, 2014). Given that numerous public comments can fully reflect functional outcomes of CLLs, which always consist of positive and negative evaluations; and kinds of efficacy/restriction factors in CLL function implementation are widely mentioned in plenty of public positive/negative comments. Meanwhile, the related research suggests that extracting themes from adequate qualitative materials is an important method to select evaluation indicators, which can reduce the potential bias in selecting indicators (Gunnarsdóttir, 2021). Numerous public reviews are important qualitative materials, offered by different individuals in different times, being relatively objective. Therefore, this study extracted positive (efficacy) / negative (restriction) indicators from public positive / negative comments (in 102 online commentary articles related to the word language of CLL) on the CLL functional outcomes. (the corresponding contents lies in lines from 512 to 522, and from 707 to 709).

Fig. the functional indicators structure of CLLs

(2) Throughout, running a recreational zone involves 3 actions: space construction, recreation experience, and marketing. This provides a theoretical basis for us to extract themes and categories from the qualitative materials. Correspondingly, 3 main categories had been exactly summarized (as the above figure shown), which include 12 sub-categories (including positive aspects and negative aspects). The 12 sub-categories served as functional indicators of CLLs in this study, and the three main categories served as their functional dimensions. (the corresponding contents lies in lines from 713 to 720).

Second, we further explained for the marketing function of CLLs, there are mainly 2 functional indicators: promoting online marketing, attracting attention on-site.

For marketing, only when the target individuals catch sight of the CLL, it can play marketing functions. Nowadays, individuals mainly find the CLL online (in travel notes, propaganda, travel tips, among others) or on site, so, the marketing function of CLLs can also mainly be evaluated from 2 aspects: promoting online marketing, and attracting attention on site. And the qualitative material also mainly reflects these 2 points. (the corresponding contents lies in lines from 715 to 720).

Reviewer Comment 2: Section 5.1 of the paper lacks depth and fails to highlight the main focus of the discussion. To improve the quality of the paper, it is recommended to emphasize the differences between this study and previous research and to clearly articulate the innovative contributions of this study.

Response: thanks for your comments, according to which, we noticed that we needed to rewrite the discussion part. Therefore, we made this part more detailed through rewriting and enlarging. The words had been increased from 308 to 1029. Following your advices, we made our effort to emphasize the similarities and differences between this study and previous research, and articulate the contributions of our study to peoples’ knowledge on CLL functions and the CLL installation practices in recreational zones. We request your review. (the corresponding contents lies in lines from 834 to 929).

Reviewer Comment 3: The authors should carefully read over their manuscript. Spelling and grammar errors should be corrected. The review of a native speaker or language editing services would be helpful.

Response: thanks for your important advice. After this revision, we invited one of our colleagues, who had ever been in England for 6 years and had published 5 papers in English, to check and revise the spelling and grammar errors. Meanwhile, we planned to get a further English language editing services from Editage (a company), which assigned a native speaker to edite our English writing in our former manuscript, if the English language need further revision and editing.

 

Response to third peer review feedback

Dear expert:

Thank you for your approval and giving us the opportunity to revise our manuscript. Your advices are thoughtful and valuable, and greatly aids us in revising the paper, as well as the important guiding significance to our researchers. After fully studying your comments, we made a careful and diligent revision which we hope meet with approval. To facilitate your review, we highlighted the primary revisions in track changes mode, and in the response, we also noted down the line numbers where the revision had been made. The main revisions are as follows. We kindly request your review.

Reviewer Comment 1: Problem formulation is quite vague.

Response: thanks for your comments. Accordingly, we added the further statements mainly in the “introduction” part, and rewrite some contents of the introduction. And for this, the introduction part was enlarged from 405words to 664 words. In order to make further explanation, 10 more literatures were referred. The corresponding statement includes the following key points.

The LL is the important semiotic resource in recreational zones, and its functions have been revealed by many relevant researches; the CLL is a kind of special semiotic resource as creativity is regarded as a key tool for promoting tourists’ experiences in many recreational zones, and in practice, many operators attached importance to the CLL, and some of them consider setting up CLLs as a crucial strategy for attracting visitors; however, seldom research explores what functions the CLLs play in recreational zones on earth; in view of this, this study attempts to answer the following related questions that are not fully discussed in existed researches: (1) What functions do CLLs have in recreational zones? (2) In practice, some CLLs have also drawn criticism, so, how to evaluate CLL functions roundly? (3) given that lot of language signs bring confusions to tourists, and how to reasonably and effectively configurate CLLs? (the corresponding contents lies in lines from 26 to 118).

Reviewer Comment 2: Function distinctions of the CCL must be presented explicitly. It is not clear enough why we have these nine groups?

Response: thanks for your constructive suggestion. After your suggestion, we made the following revision.

According to relevant research (Pléh, 2018), the language functions in two aspects: it mainly delivers “deictic & exact” information, or “emotional & abstract” information (the corresponding contents lies in lines from 442 to 444). Moreover, through literature review, we concluded that the specific functions of the CLLs included: improving spatial expressions, enhancing tourists’ experiences (eliciting visitors’ feelings and bringing them deeper experiences), and stimulating and regulating tourists’ behavior (the corresponding contents lies in lines from 305 to 343).

So, taking the above two perspectives into account comprehensively, we identified that in delivering “deictic & exact” information, what types of functions the CLLs played respectively in the aspects of improving spatial expressions, enhancing tourists’ experiences and stimulating tourist behaviors; and meanwhile identified that in delivering “emotional & abstract” information, what types of functions the CLLs played respectively in the above 3 aspects. Consequently, we summarized 9 function types of CLL, as shown in the following table. (the corresponding contents lies in lines from 495 to 502, and from 648 to 655).

Table. The function types of CLL

perspectives delivering deictic & exact information delivering emotional & abstract information

improving spatial expressions S7, S8 S2

enhancing tourists’ experiences S1, S3, S4, S5, S9

stimulating tourist behaviors S6 

Reviewer Comment 3: Not all links work properly shown in the table 1. It should be useful to present the images of all types of the CCL.

Response: thanks for your reminding. The revisions that we made are as follows.

First, we changed some CLL example pictures, and spontaneously, the corresponding links were also changed. Considering that, perhaps some links can not work in other state, we provided the screenshot of the webpages on which the corresponding example pictures were gotten (as shown in the appendix behind this response).

Second, after your advice, we presented the images of all types of the CLL, as following.

Reviewer Comment 4: The information concerning function indicators must be structured showing the reasoning of the chosen characteristics from the theoretical and implementational views of the point.

Response: thanks for your commentary. The revisions and further explanations that we made are as following.

We further stated the reason and procedure for choosing the function indicators (the corresponding contents lies in lines from 464 to 468, from 512 to 522, from 707 to 709, and from 713 to 720), the main points including: 

First, efficacy factors and restriction factors both influence the implementation of CLL functions (Wang, 2014). (the corresponding contents lies in lines from 512 to 513, and from 707 to 709.)

Second, numerous public comments on CLLs always consist of positive and negative evaluations, and kinds of efficacy/restriction factors in CLL function implementation are widely mentioned in these comments. Meanwhile, previous research believed that extracting themes from adequate qualitative materials can reduce the potential bias in selecting indicators (Gunnarsdóttir, 2021). (the corresponding contents lies in lines from 514 to 522)

Third, Throughout, running a recreational zone involves 3 actions: space construction, recreation experience, and marketing. (the corresponding contents lies in lines from 713 to 720)

Therefore, based on the qualitative materials, considering efficacy and restriction factors, we identified that in the above 3 actions, what factors positively or negatively influence the function implementation of CLLs. Accordingly, an indicator’s structure was constructed, as shown by the following figure. Additionally, in this revision, we also found some former expression of main function category was inappropriate, and we changed it.

Fig. the functional indicators structure of CLLs

Reviewer Comment 5: All these parts of the text must be moved to theoretical part of the manuscript.

Response: thanks for your comments. As you mentioned above, identifying function types of CLL and choosing the function indicators for evaluating the CLL leisure functions are key points in our study, for making our addressing means clearer, we constructed a theoretical framework following your comment.

First, the language functions in two aspects: delivering deictic & exact information, or delivering emotional & abstract information (Pléh, 2018). This study attempted to find out what functions would be played during the CLLs delivering deictic & exact information, and what functions would be played during they delivering emotional & abstract information. (the corresponding contents lies in lines from 438 to 450)

Second, according to previous research (Bugental, 1966), the accomplishment of the functions of CLLs involves the CLL setters, readers, and the related objects of the recreational zones. Based on the objects, the setter needs to accomplish expressions, and the reader needs to be effectively impressed. So, for evaluating the CLL functions, we attempted to explore whether the setters’ expressions were positively or negatively influenced by some factors related with the CLLs, and whether the readers’ experiences were positively or negatively influenced. If more positive influence factors exist, the CLLs will play the better functions, and vice versa. Naturally, in the functional implementation of CLLs, both efficacy and restriction factors work. Therefore, this study discerned these related efficacy and restriction factors in the function implementation of CLLs, and accordingly evaluated the leisure functions of different CLLs. (the corresponding contents lies in lines from 451 to 468)

Third, based on the above 2 aspects, we constructed a theoretical framework, on which the inquiry and analysis are conducted. It can be described as follows. Under the object condition of specific recreational zone, the setter can efficiently express the leisure elements with CLLs; In setting up CLLs, some efficacy and restriction factors will influence the implementation of the CLL functions; Through transmitting “deictic & exact” or “emotional & abstract” information, the installed CLLs appeal to the readers among tourists; Consequently, CLLs contribute to improving spatial expression, and promoting the readers’ tourism experiences and behaviors. (the corresponding contents lies in lines from 469 to 476)

The following figure shows the theoretical framework.

Fig. the theoretical framework

Reviewer Comment 6: How the weights presented in table 2 are calculated?

Response: thanks for your comments. Accordingly, from the following aspects, we made more explanations: (1) we calculated the weights with the entropy-weighted method, for this method, an evaluation matrix was required. Therefore, we made more detailed explanation on how this matrix had been made. (2) we made more detailed explanation on how the entropy-weighted method was applied and proceeded. We request your check. (the corresponding contents lies in lines from 579 to 606)

Reviewer Comment 7: The need and the analysis of the results in the table 3 must be clarified.

Response: thanks for your comments. We make more explanation and clarification for this. The key points are as follows. For further analysis, the data from ranking need to meet the 2 requirements: (1) At specific aspect, different types of CLL reveal different leisure function outcomes; this can be checked with Friedman test, and if the Friedman statistic is more than the critical value with a higher significance level, the test is passed; (2) At specific aspect, the function outcome ranking of various types of CLL made by different participants need show some consistency; this can be checked with Kendall test, and a concordance coefficient more than 0.6 at a high significance level implies a stronger consistency (Gearhart, 2013) of the ranking results given by different individuals. Table 3 shows that the data passes the Friedman test and Kendall test, and is qualified. (the corresponding contents lies in lines from 748 to 753, from 757 to 758, from 761 to 763, and from 766 to 767)

Reviewer Comment 8: What metrics are applied to evaluate the considered cases of the CCL?

Response: thanks for your enlightening question. We think this mainly involves 2 issues that need to be further clarified: the construction of the evaluation indicator system, and the calculation of evaluation results. (1) for the first issue, we made extra detailed explanation in the part of “3.3 summarizing CLL function indicators” (the corresponding contents lies in lines from 512 to 522), and in the part of “4.2 Function Indicators of CLLs” (the corresponding contents lies in lines from 707 to 722). With the indicator system, we obtained the evaluation data through respondents ranking the functional outcomes of various types of CLL for each function indicator; and then, the ranking results were transformed into function evaluation values ranging from 1 to 9. (2) for the second issue, also according to your suggestion on the analysis method, we used a fuzzy PROMETHEE method to proceed our analysis and got the final evaluation results on the leisure function outcomes of different types of CLL. (the corresponding contents lies in lines from 610 to 643, from 792 to 793, and lies in lines 810 and 828) We request your check.

Reviewer Comment 9: How the vagueness of the initial information has been taken inti account? Nowadays the different fuzzy approaches have been proposed to consider the uncertainty of the initial information. Therefore, there is no any novelty concerning the solution method of this decision making problem.

Response: thanks for your constructive suggestion. The TOPSIS method previously used in our study indeed didn’t address the vagueness of the initial information. For this, we proceeded a further study for fuzzy approach, and consulted other researchers to find solution. And we found that the fuzzy PROMETHEE approach could be used in our study. Based on the fuzzy judgment matrix, with this approach, the probability that one item is superior to another would be judged through preference function, and then the fuzzy decision would be achieved. Although the ranking results are almost as same as the result in previous analysis (in some function dimension, the ranking results have some differences), there are many nuances. And we think the newly used approach is more persuasive. (the corresponding contents lies in lines from 610 to 643, from 792 to 793, and lies in lines 810 and 828) We request your check.

 

Response to Journal Requirement

Dear Editorial Team,

Thanks for your proceeding our manuscript. Your requirements and comments are very imprtant to improve our manuscript. We carefully studied these requirements and advices, and make our effort to meet them. We kindly request your checking; if some problems still remains, we will furtherly address them until our manyscript meet your standards.

1. To ensure that your manuscript meets PLOS ONE's style requirements, including those for file naming.

Response: we checked our manuscript according to the requirements.

2. To provide additional details regarding participant consent. In the ethics statement in the Methods and online submission information, please ensure that you have specified what type you obtained (for instance, written or verbal, and if verbal, how it was documented and witnessed). If your study included minors, state whether you obtained consent from parents or guardians. If the need for consent was waived by the ethics committee, please include this information.

Response: we added corresponding statements in our manuscript, and kindly request your check. (The corresponding contents lies in lines from 553 to 557) 

Response: we don’t wish to make change in this aspect.

4. To include your full ethics statement in the ‘Methods’ section of your manuscript file. In your statement, please include the full name of the IRB or ethics committee who approved or waived your study, as well as whether or not you obtained informed written or verbal consent. If consent was waived for your study, please include this information in your statement as well.

Response: we wrote the full name of the ethics committee who approved our study. And we wrote that we had obtained written informed consent. (The corresponding contents lies in lines from 553 to 557)

5. To include captions for your Supporting Information files at the end of your manuscript, and update any in-text citations to match accordingly.

Response: we have included the captions of our supporting Information files at the end of our manuscript, and there are no corresponding in-text citations.

 

Appendix: Screenshot of online search results for the example pictures of CLLs

(1) For emotional arousal

(2) For creating atmosphere

(3) For mood regulation

(4) For mentality guidance

(5) For pandering to wishes

(6) For inducing behaviors

(7) For labeling site

(8) For building cognition

(9) For resonating with movie lines

---

## [Decision Letter · Decision Letter 1]

16 Feb 2024

Promoting leisure functions through setting creative linguistic landscapes in recreational zones

PONE-D-23-26514R1

Dear Dr. Tian,

We’re pleased to inform you that your manuscript has been judged scientifically suitable for publication and will be formally accepted for publication once it meets all outstanding technical requirements.

Kind regards,

Academic Editor

PLOS ONE

Additional Editor Comments (optional):

Reviewers' comments:

Reviewer's Responses to Questions

**Comments to the Author**

1. If the authors have adequately addressed your comments raised in a previous round of review and you feel that this manuscript is now acceptable for publication, you may indicate that here to bypass the “Comments to the Author” section, enter your conflict of interest statement in the “Confidential to Editor” section, and submit your "Accept" recommendation.

Reviewer #1: All comments have been addressed

Reviewer #4: All comments have been addressed

2. Is the manuscript technically sound, and do the data support the conclusions?

Reviewer #1: Yes

Reviewer #4: Yes

3. Has the statistical analysis been performed appropriately and rigorously? 

Reviewer #1: Yes

Reviewer #4: Yes

4. Have the authors made all data underlying the findings in their manuscript fully available?

Reviewer #1: Yes

Reviewer #4: Yes

5. Is the manuscript presented in an intelligible fashion and written in standard English?

Reviewer #1: Yes

Reviewer #4: Yes

6. Review Comments to the Author

Reviewer #1: Please consider adding further references.

E.g.

https://www.tandfonline.com/doi/full/10.1080/23247797.2021.2014029

and

Alhendi Osama, Dénes Dávid Lóránt, Fodor Gyula, Fredrick Collins Adol Gogo, Balogh Péter

The impact of language and quality education on regional and economic development: a study of 99 countries

REGIONAL STATISTICS 11 : 1 pp. 42-57. , 16 p. (2021)

https://www.ksh.hu/statszemle_archive/regstat/2021/2021_01/rs110101.pdf

Etc.

Reviewer #4: Many thanks, you have adequately addressed your comments raised in a previous round of review and weel done.

7. PLOS authors have the option to publish the peer review history of their article (what does this mean?). If published, this will include your full peer review and any attached files.

Reviewer #1: No

Reviewer #4: No

---

## [Editor Report · Acceptance letter]

13 Mar 2024

PONE-D-23-26514R1 

PLOS ONE

Dear Dr. Tian, 

I'm pleased to inform you that your manuscript has been deemed suitable for publication in PLOS ONE. Congratulations! Your manuscript is now being handed over to our production team.

Kind regards, 

on behalf of

Mr Youssef El Archi 

Academic Editor

PLOS ONE